# Patient and Public Involvement in Sexual and Reproductive Health: Time to Properly Integrate Citizen’s Input into Science

**DOI:** 10.3390/ijerph17218048

**Published:** 2020-10-31

**Authors:** Miguel García-Martín, Carmen Amezcua-Prieto, Bassel H Al Wattar, Jan Stener Jørgensen, Aurora Bueno-Cavanillas, Khalid Saeed Khan

**Affiliations:** 1Department of Preventive Medicine and Public Health, Faculty of Medicine, University of Granada, 18016 Granada, Spain; mgar@ugr.es (M.G.-M.); abueno@ugr.es (A.B.-C.); profkkhan@gmail.com (K.S.K.); 2Consortium for Biomedical Research in Epidemiology and Public Health (CIBERESP), 28029 Madrid, Spain; 3Instituto de Investigación Biosanitaria (ibs.Granada), 18014 Granada, Spain; 4Reproductive Medicine Unit, Institute for Women’s Health, University College London Hospitals, London WC1E 6BT, UK; dr.basselwa@gmail.com; 5Warwick Medical School, University of Warwick, Coventry CV4 7 AL, UK; 6Department of Obstetrics and Gynaecology CIMT-Centre for Innovative Medical Technologies Odense University Hospital, 5000 Odense C, Denmark; jan.stener@rsyd.dk

**Keywords:** patient and public involvement, women, health, research, international collaboration

## Abstract

Evidence-based sexual and reproductive health is a global endeavor without borders. Inter-sectorial collaboration is essential for identifying and addressing gaps in evidence. Health research funders and regulators are promoting patient and public involvement in research, but there is a lack of quality tools for involving patients. Partnerships with patients are necessary to produce and promote robust, relevant and timely research. Without the active participation of women as stakeholders, not just as research subjects, the societal benefits of research cannot be realized. Creating and developing platforms and opportunities for public involvement in sexual and reproductive health research should be a key international objective. Cooperation between healthcare professionals, academic institutions and the community is essential to promote quality research and significant developments in women’s health. This cooperation will be improved when involvement of citizens in the research process becomes standard.

## 1. Introduction

Evidence-based sexual and reproductive health requires a global effort, with political and societal actions. A comprehensive paradigm shift is needed, where not only health professionals and researchers are ready to implement citizen participation, but also where political and administrative control of the health system allows and supports it [1]. Public and patient involvement is a field where policy has tended to outpace evidence. Exploring the impact of citizens in research is limited to investigating researchers’ and public and patient contributors’ reports of their views and experiences. Objective techniques for evaluating the impact and its influences remain hard to reach in a process that is inherently relational, subjective and socially constructed [2]. Public and patient involvement is as an expression of a democratization of healthcare, and a political and managerial tool to ensure quality, documentation, and equal treatment, thereby governing and controlling a public healthcare system. For example, the Danish public healthcare system is increasingly influenced by politicians. The use of standardized schemes and checklists used for documentation and quality assurance underlines this development, and patient and public involvement might be seen as adding to this movement [1].

The future worldwide vision is focused on building healthier lives, driven by the evidence-based provision of evidence-based healthcare to improve citizen’s quality of life [3] However, evidence-based medicine faces challenges due to: lack of proper patient and public involvement or public and participant involvement (PPI) in research, leading to poor buy-in [4]; lack of incorporation of patients’ lived experiences in research outcomes, reducing the relevance of evidence [5]; and lack of recommendations for shared decision-making in guidelines leading to ineffective communication between health workers and patients [6].

In the last decade, the incorporation of public participation initiatives has shown the potential to improve the quality and the object of research, allow a wide dissemination of the findings found, and promote a better integration of the results in clinical practice [7,8,9]. Public participation also allows an approach to sensitive phenomena such as sexual and reproductive health, or ethical aspects that arise during the development of the study [7].

Significant development of women’s health requires international collaborations and participation of women in the research process. To improve sexual and reproductive health research, our objective should be to establish the importance of creating and developing the platforms and tools for public involvement. In this sense, creation of new tools and platforms involving a combination of traditional outreach and online strategies. Face-to-face invitations, training meetings, local civil society organizations (CSO) and family medicine health center support can be combined with online tools, such as online announcements, use of google meet, hangout or zoom, and social media platforms (Facebook, Twitter). These tools offer opportunities for community members to engage in interactive ways and can bring new input from the patient and participant involvement process.

## 2. Evidence-Based Sexual and Reproductive Health

Reproductive health (RH) was defined in 1994 at the International Conference on Population and Development as a “state of complete physical, mental and social well-being and not merely the absence of disease or infirmity, in all matters relating to the reproductive system, and to its functions and processes” [7]. Sexual and reproductive health (SRH) includes a number of health problems, such as (1) family planning; (2) maternal and newborn health care; (3) prevention, diagnosis and treatment of sexually transmitted infections (STIs) [8]; (4) adolescent sexual and reproductive health; (5) cervical cancer detection; and (6) prevention and management of infertility [9]. The World Health Organization include the Sexual and reproductive health in the health topics and highlighted that ‘Sexual and reproductive health is a very personal subject, so people may have trouble finding or asking for accurate information about it. This may also help explain why these issues are still not addressed openly, and services are inadequate, fragmented and unfriendly in some countries in the European Region’ [10]. Sexual and reproductive health services aim to prevent poor sexual and reproductive health, such as complications of pregnancy and childbirth, unwanted pregnancies, unsafe abortions, complications caused by STIs, sexual violence, and women dying from preventable cancer [10].

Reproductive health, including sexual health, and reproductive rights, as well as gender equality and women’s empowerment, are relevant to improving the quality of life for everyone [7] as affirmed at various international forums, such as the International Conference on Population and Development (ICPD) and The Fourth World Conference on Women: Action for Equality, Development and Peace [11,12]. The Millennium Development Goals related to maternal and child health have still not been achieved in many countries; considering 35 countries, less than 30 percent of women of reproductive age use modern contraceptive methods. The method choice is still limited in many countries, due to lack of access or provider biases. Although there are good options for safe abortion, these services remain unavailable in many countries due to legal barriers, lack of training, and stigma. However, significant progress has been made in improving reproductive health. For example, family planning has expanded around the world, spurring a broader coverage of services with greater emphasis on quality and human rights; adolescent sexual health has been addressed with effective messages and services and new approaches try to reduce gender-based violence and clinical and policy guidelines have been incorporated [11].

The 2020 Sustainable Development Goals Report in goal five (gender equality) highlighting that there is still a lack of decision-making power for women, extending to their own reproductive health and that slightly more than half of all women (55%) make their own decisions when it comes to sexual and reproductive health and rights, based on 2007–2018 data from 57 countries on women aged 15 to 49, who are either married or in union. The analysis also found that only three in four women are making their own decisions regarding health care or on whether or not to have sex [12]. United Nations Member States have moved on to confirm their commitment to Sustainable Development Goals (SDG) including its targets designed to “ensure universal access to sexual and reproductive health (SRH) care services, including for family planning, information and education, and the integration of reproductive health into national strategies and programs by 2030” and to “ensure universal access to sexual and reproductive health and reproductive rights” [9].

Patient and public involvement in research could be an approach to effectively address poor sexual and reproductive health outcomes among women [13]. This enables researchers to identify better strategies, focusing, for example, on culturally and contextually appropriate research and prevention, equitable access to effective sexual health information, or quality education and training for public health professionals. The health, demographic change and well-being programs [14] are committed to the implementation of research to improve maternal and child health. Public participation and involvement in the investigation of pregnancy and childbirth differ from those under chronic diseases, especially because motherhood is a transitory health experience in the life of women and their families [4]. Women can have a variety of experiences in their pregnancy and delivery, ranging from low-risk, home birth without intervention to prolonged hospitalization with long-lasting morbidity [15]. Their own experience will influence your attitudes and inevitably your contribution to research. Their participation should guide the priority areas for users of health services. Moreover, the Faculty of Sexual and Reproductive Healthcare of the Royal College of Obstetrician and Gynaecologists values patient and public involvement in the development of clinical guidelines. It actively considers patient and public involvement across all stages of the guideline development process, from consulting with individuals on the proposed scope of the clinical guideline through to getting feedback on patient summaries. In order to facilitate patient and public involvement, a variety of methodologies (e.g., questionnaires, focus groups, interviews) across different settings (e.g., clinics, online, community) are considered [16].

## 3. Patient and Public Involvement

Effective patient and public involvement can be defined as where research is “being carried out “with” or ”by” members of the public” not just ”to”, ”about” or ”for” them” [17]. The involvement process needs to change from an ad-hoc informal consultation into an established partnership underpinned by dedicated systems and infrastructure [18]. It must be implemented throughout the research lifecycle from idea generation, prioritization and commissioning into research conduct, publication and implementation in clinical practice [4,19].

It has been known for some time now that patient and public involvement has an important role in the identification of specific research questions [20,21], and that subsequent requests for research funding are more successful. Furthermore, it may be that public participation reverses in the reformulation of research questions [22], or even in the abandonment of a research project request if the involvement suggests that the research questions are not meaningful to the patient, public or citizens [23]. Additionally, patient and public involvement improves trial design [24], by ensuring that the design is acceptable to potential participants [25], facilitates recruitment for studies [26] and allows access to underserved and rarely heard populations, and patient and public involvement helps to disseminate the research results [27]. In short, patient and public involvement increases researchers’ confidence in their studies [24] and promotes a greater understanding of the patient’s perspective [28].

Internationally, there is increasing recognition of the importance of involving patients and the public in health-care governance and research [29]. Policymakers give the relevance to patient and public involvement as characterized by the phrase ‘nothing about me without me’ [30]. Thus, local [31] and international organizations, such as ‘The European Patient’s Forum’ or ‘The International Association for Public Participation’ [32,33] are beginning to harness the power of participation by health service users, promoting public participation in research planning and the development of resources and information to support citizen participation and patient engagement. Promoting the role of patients in health research lifecycle has been at the heart of various international efforts including the European Network for Health Technology Assessment [34] and the Innovative Medicines Initiative which have called for better patient and public involvement in research synthesis projects across Europe [35]. Recently, the European Patients’ Academy (EUPATI), a pan-European project was established as a collaborative public–private multi-stakeholder partnership of pharmaceutical industry, academia, and not-for-profit patient organizations [36]. These initiatives have developed various tools, platforms, and training resources for patient advocates to improve their participation in research with focus on four areas: pharmaceutical industry-led medical research, ethics committees, regulatory authorities, and health technology assessment. In turn, prestigious journals such as the British Medical Journal (BMJ) have also emphasized the use of patients and citizens as editors and emphasize the need for patients to participate as co-authors in the articles that are published [37].

Involving lay volunteers for problem-solving provided insights enhanced research design and served to identify weaknesses and barriers. Tensions and barriers have been generated with patient and public involvement in research [38]. On one hand, shared tensions usually are due to unclear roles, absent reporting guidelines, exclusion, framework limitations, resource allocation, and administrative boundaries. For example: active public involvement in the decision-making process of designing trials is less common than consultation on what has already been decided. Volunteers report needing early involvement to propose constructive changes. Researchers’ worries about aggressive patients and those without respect for rules of confidentiality or data protection held up the research. Researchers may feel overwhelmed when they cannot fulfill the expectations they have over patients and public involvement [39]. On the other hand, shared barriers include those imposed by cultures, values, and power hierarchies. Limited involvement of the health community may occur through coalitions, collaborations, and partnerships [38].

Because research literacy could be an inconvenience in the process of patients and public involvement, some suggestions for getting the best from public involvement have been published [38]: ongoing support and implementation; training/capacity building; inclusion process; building trust and community and reinforcing patients’ value and validity. In the second suggestion, capacity building, it is necessary to (1) provide training in research literacy and ethics, drawing on the many training programs that are available, (2) at every meeting have a jargon bin, when an unfamiliar term comes up, define it and use this to build glossaries, (3) promote a reciprocal learning relationship, letting volunteers know that researchers have made a long-term commitment to the patient and public partnership in research, and (4) encourage realistic expectations in volunteers and researchers and manage relationships with respect. Moreover, the use of patient and public involvement in dissemination planning, design, implementation, and distribution could increase public involvement, contribute to health literacy, and expand knowledge for patient values and preferences. The addition of patient reviewers by journals may contribute to health literacy and provide insights for future participatory research practice [38].

## 4. Public Involvement in Research to Address Gaps in Evidence

There is growing recognition of the importance of experiential knowledge being approached alongside scientific knowledge [40]. Patient and public involvement throughout the research lifecycle and in research syntheses, such as systematic reviews and clinical practice guidelines, is being demanded by funders, but there is often an intrinsic resistance in the field of biomedical research [41]. For example, the chief investigator who expressed skepticism about patient and public involvement, focused mainly on using public engagement to meet funding requirements or by including them on trial steering committees, whereas those who valued patient and public involvement often described in detail how it was of benefit within their trials [2]. Translation and effective implementation of evidence into practice through requires careful planning in consultation with patients as key stakeholders. When involving patients, it is imperative to take the perspective of the patient and their way of life as a starting point. Patient participation is achieved by inviting them to participate in their treatment and care, and in the research that surrounds them. It requires health professionals to be involved in the daily life of their patients, but it also requires organization and shared decisions that promote patient participation in daily clinical practice [1]. Then, establishing patients as a stakeholder group in a true partnership with researchers is the key to addressing these challenges [42,43]. Health research funders and regulators are promoting patient and public involvement in research and research syntheses [29]. However, many clinicians, researchers, systematic reviewers and guideline makers still do not have the tools required to involve patients. It is not always recognized that through their lived experiences and social background, patients can positively be involved in research with their unique perspectives, increasing its impact on society [44].

Another aspect of interest that must be taken into account is that, in the past, the choice of results in research studies was driven by researchers who did not consider the participation of patients or citizens [6]. This practice has often led to heterogeneity, making it difficult to synthesize research, replicating the same studies, and increasing research waste [5]. The implementation of core (standardized) sets of important outcomes for each health condition has been suggested as a possible novel solution to this problem [45]. A core outcome set (COS) is a minimum consensus-based set of results that should be measured and reported in all clinical trials for a specific health condition or intervention [46]. Issues related to outcome assessments, such as selective reporting and inconsistency between studies, can be addressed by developing a core set. Development requires reaching a consensus on: (1) core outcome domains and (2) core outcome measurement instruments. Besides, methods used to reach consensus include systematic reviews of the literature to inform the process, qualitative research with physicians and patients, group discussions (for example, using the nominal group technique), and structured surveys (for example, using the Delphi technique). Multiple stakeholders must be involved in the process, and particular attention must be paid to patients [47]. Traditionally, the guidelines promoted by leading health experts have generated subjective recommendations that correspond to the provision of local, regional or national clinical care [48]. This has produced a limited effect on health outcomes and has promoted high and unjustified variation in the provision of care and services to users.

We have to adopt a novel approach by involving a wide range of key stakeholders of different disciplines with direct involvement in healthcare planning and provision for women and children. A strong level of cooperation is necessary via effective communication channels between healthcare professionals, academic institutions and the community [49]. Stakeholders and activities for participant involvement in research are shown in Figure 1. We suggest that these developments include clinicians (e.g., gynaecologists and paediatricians), methodologists (e.g., statisticians and systematic reviewers) and lay consumers (e.g., patients with lived experiences and civil society organization (CSO) representatives and related to sexual and reproductive health). Clinicians and professional bodies should promote patient and public participation in research helping to develop know-how in participants engagement. Methodologists—researchers, guideline makers and systematic reviewers included—will help in the scientific training of lay research enablers, to develop the local know–how on participant involvement in research. Lay consumers will provide the voice of lay sexual and reproductive health participants of all backgrounds to engage in the research process and to be involved in the dissemination of the research. The patient perspective is valuable to ensure quality in healthcare. However, the patient perspective is only one perspective, similar to the ones of other important groups (e.g., health professionals, leaders, administrators, politicians). A selected coordinator for patient and public involvement is key to facilitating interactions. We agree with other authors, the coordinator should be experienced in the potential challenges of public and patient involvement, including power and control issues, and the consequences of public and patient contributors lacking knowledge of research processes, terminology, and ethical constraints [39].

Furthermore, patient and public involvement recruitment strategies are necessary, ensuring that we apply fair and transparent processes, that there is a shared understanding of the roles, offering support and learning for public participation in the research. For example: (1) women participating in actives randomized controlled trials; (2) women and men contacted from midwives who collaborate closely with the clinical institutions; (3) women and men with a disability or belonging to disadvantaged social classes; (4) social media, web sites and banners in health centers and nurseries to recruit participants and (5) civil society organizations (CSO).

The patient and public involvement should be made up of citizens and the public. They should receive advanced training on citizen participation in research, such as has been carried out within individual projects [4]. This training should include some important topics: (1) Introduction to health research; (2) Different types of research and study designs; (3) Perspective of the research from the citizen’s point of view; (4) Research life cycle; (5) Different roles for the audience (participation, engagement); (6) Why involve the public in research (including diversity and inclusion in research); (7) Case studies: examples and public impact on research; (8) How to do research with the commitment of citizens; (9) How can the public do “citizen science”?; (10) How to maximize the impact of ”citizen science”; (11) Solution of practical problems in the field; (12) Research dissemination and beyond; and (13) Measurement of the value and impact of citizen research. Developing these initiatives further, more citizens should receive basic training in citizen participation and the relevance of the investigation through dedicated training sessions. Patient and public involvement aims are varied depending on the research subject. Therefore, it is less common to involve patients and the public in the research process, than in other areas such as genetics, biobank research, cancer research, clinical trial research or decision-making [50]. The methodology mainly used for patient and public involvement in researching is information sharing, questionnaires, interviews or focus groups. Importantly, conducting a focus group at the beginning of the study and another at the end, you can identify the needs, barriers and feedback of the patient and public participations, trying to respond to how diversity impacts innovation, how to act to promote the inclusion of people with disabilities or from more disadvantaged communities, relationships between citizens, researchers and institutions, the type of language used to communicate and whether there have been unexpected opportunities. On the other hand, the research team is focused on the public and patient involvement of consultation, deliberation and participation [50].

To achieve quality results in the process of patient and public involvement in research, the Guidance for Reporting Involvement of Patients and the Public (GRIPP2) can be useful, the first international evidence-based, consensus informed guidance for reporting patient and public involvement in research [51]. There is a short and a long version of GRIPP2, that aims to improve the quality, transparency of the international patient and public involvement. The short form has five sections or topics: (1) Aim of patient and public involvement in the study; (2) Methods used in the public involvement in the study; (3) Study results: positive and negative; (4) Discussion, describing the positive and negative effects on the extent to which patient and public involvement influenced the study, and conclusions; and (5) Reflections/critical perspective, comment critically the things that went well and those that did not. On the other hand, the long form includes eight sections: (1) Abstract of the paper; (2) Background to the paper; (3) Aims of the paper; (4) Methods of the paper; (5) Measurement of the patient and public involvement: methods used to qualitatively and quantitatively measure or assess the impact of public involvement and methods used to capture or measure its impact; (6) Economic assessment of public involvement in the study; (7) Study results; and (8) Discussion and conclusions. This last section incorporates the reflections and critical perspective of the study. In order to improve current public involvement practice in the sexual health field, a recently published study carried out audits on patients and public involvement plans in the local sexual health service (*n* = 18), and a refined audit tool was completed in research projects (*n* = 5). The responses of the tools used in the audit showed a wide variability in practice. Problems included the combination of patient participation work and qualitative research, lack of goals in patient participation in research and lack of responsiveness around patient needs, and insufficient resources to work with patient and public involvement. Specific research problems included late participation of patients after key decisions had already been made [52].

At present, there are still some limitations about the evidence of the patient and public involvement research outcomes. In a systematic review, mapping the impact of the patient and public involvement in health and social care research [42], authors aimed to examine the conceptualization, measurement, impact and outcomes of patient and public involvement in health and social care research. This study provides the first international evidence of patient and public involvement impact that has emerged in the research process, however, much of the evidence concerning the impact remains weak.

It is important to remark that the “ReseArch with Patient and Public invOlvement: a RealisT evaluation” study (RAPPORT study) [53] aimed to determine the types of patient and public involvement in funded research, analyze the contextual and dynamics of patient and public involvement and explore the experience of patient and public involvement in research. They found that six striking actions were required for effective patient and public involvement: (1) A clear purpose, role and structure for patient and public involvement; (2) a key individual co-ordinating patient and public involvement ensuring diversity; (3) whole research team engagement with patient and public involvement, this is a research team positive about patient and public involvement input and engaged with it; (4) mutual understanding and trust between the researchers and lay representatives; (5) ensuring opportunities for patient and public involvement throughout the research process; and (6) and evaluating patient and public involvement in a proactive and systematic approach. Future work is required, exploring the impact of virtual patient and public involvement, economic evaluation of patient and public involvement, and a longer-term follow-up study of the outcomes of patient and public involvement on research findings and impact on services and clinical practice.

## 5. Clinical Relevance and Goals of Patient and Public Involvement in Research

The most documented benefits of patient and public involvement in research are the involvement of the patient and public groups at all stages of the trial [39], utility in randomized controlled trials [2], contribution in the design of clinical trials [38], and perceptions of patient involvement in research and clinical practice [1].

In a qualitative study of patients and researchers from a cohort of randomized trials [2], 21 chief investigators, 10 trial managers and 17 patient and public involvement contributors were interviewed from 28 trials. The conclusions were that to enhance patient and public involvement in trials is crucial to develop goals for patient and public involvement at an early stage that fits the needs of the trial, plan patient and public involvement implementation in accordance with these goals, invest in developing good relationships between patient and public involvement contributors and researchers, and favor responsive and managerial roles for contributors. A systematic overview of public and patient involvement in clinical trials design included 27 reviews [38]. Public involvement roles were primarily based on agenda setting, steering committees, ethical review, protocol development, and piloting. Public involvement was reported to have increased the quantity and quality of patient relevant priorities and outcomes, enrolment, funding, design, implementation, and dissemination.

We would like to mention a couple of examples were patients and the public were involved in research. The first is a 3D study [39], where fourteen people living with multiple long-term conditions (multimorbidity) were included as patient and public involvement contributors to a randomized controlled trial to improve care for people with multimorbidity. Meetings took place approximately four times a year throughout the trial, beginning at the grant application phase. Their experience also supports the findings of the RAPPORT study [53] in four of the six conditions most influential in establishing effective patient and public involvement: (1) a research team positive about patient and public involvement, (2) a key individual to co-ordinate, ensuring diversity, and relationships that are established and maintained over time, (3) a shared understanding of moral and methodological purposes of patient and public involvement and (4) proactive and systematic evaluation of patient and public involvement. The second example is a secondary analysis of women’s perceptions and experiences of egg aspiration in fertility treatment [1]. This secondary analysis was initially inserted in a randomized controlled study where the aim was to gain insight into perceptions and experiences within a group of women undergoing fertility treatment through two focus group interviews. They pointed out that knowledge and awareness of the difference in perspectives is important when healthcare professionals seek to involve patients both in clinical practice and in research.

Secondly, patient and public involvement in research has doubled the impact in trials: focused and diffuse [2]. Focused impact comprises patient and public contributors’ input that, from the perspective of the informant, changed or influenced an aspect of the trial, whereas diffuse impacts comprise patient and public contributions that influenced the way researchers thought or felt about the trial. Focused impacts included patient and public contributors helping to choose the primary outcome for the trial and to increase recruitment through their contacts and networks. Diffuse impacts largely entailed interactions between researchers and patients and public contributors that helped to reassure the research team and increase or maintain their confidence and motivation for the trial.

Specific examples of the impact of patient and public involvement in researchers’ professional lives are those where patients become co-authors in publications, co-applicants in projects call and trainers [4,54,55,56] or even becoming new editors in a prestigious journal, such as the BMJ Sexual and Reproductive Health [57], who wrote: ‘Patient involvement is an important piece of cultural change in healthcare, and will require commitment, time and sensitivity, and a willingness to experiment and learn from mistakes’.

## 6. Conclusions

Without the active participation of citizens, sexual and reproductive health research risks losing relevance and validity. Health research funders and regulators are promoting patient and public involvement in research but there is a lack of quality tools for involving patients. International collaborations, and the cooperation between healthcare professionals, academic institutions and the community are essential to promote quality research in sexual and reproductive women’s health. A significant development in women’s health will be achieved when the involvement of citizens in the research process becomes standard.

## Figures and Tables

**Figure 1 ijerph-17-08048-f001:**
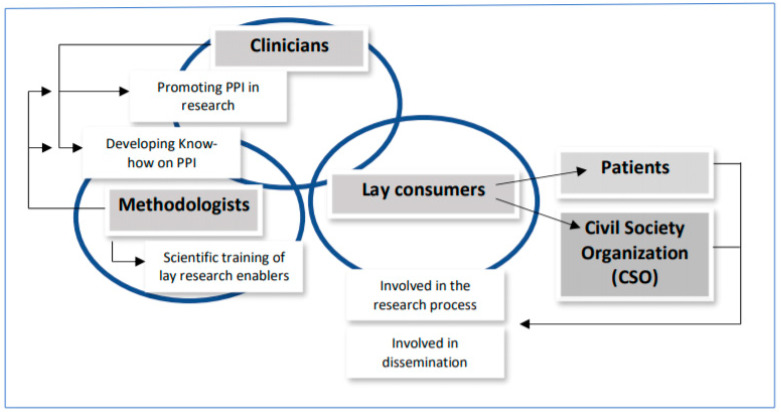
Key stakeholders and activities in public and participant involvement (PPI).

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
