# Peer review of "Patient and Public Involvement in Sexual and Reproductive Health: Time to Properly Integrate Citizen’s Input into Science"

_ijerph, 2020, doi:10.3390/ijerph17218048_

Round 1

Reviewer 1 Report

n this commentary the Author highlighted that the achievement in sexual and reproductive health is a global goal. The Authors illustrated that health research funders and regulators are promoting patient and public involvement in research but this appears difficult to obtain because there is a lack of quality tools for involving patients. The Authors sustained that the partnerships with patients are necessary to produce and promote quality research, and that without the active participation of women as stakeholders the societal benefits of research cannot be realised. The Authors identified in the Creation and develop of platforms and opportunities for public involvement in sexual and reproductive health research a key international objective. The Authors sustained also that the cooperation between healthcare professionals, academic institutions and the community is essential to promote quality research and significant development in women's health will be achieved when involving of citizens in the research process becomes standard. This commentary is within the remit of International Journal of Environmental Research and Public Health, section Women’s Health, and it should be revised before it can be accepted for publication.

This is a commentary and not an original study or research study and for this reason I have only some comments or considerations to report:

  • The first sentence in the abstract is the same of the introduction, I suggest to rephrase one of them, it is not common to see the same phrase in different sections.
  • The Authors correctly highlight that as regards sexual and reproductive health there are still many goals to obtain also in the research area, and the PPI may be a correct  strategy. However, I suggest to deepen the sections related: to the creation of new tools and platform; to the specific model proposed by the authors, and to the studies and programme that have included PPI. I also suggest to deeply illustrate both the clinical relevance and the goals that should be obtained through this approach.
  • Check for English language.

Author Response

Commentary or suggestion

Author´s responses and changes

1.     In this commentary the Author highlighted that the achievement in sexual and reproductive health is a global goal. The Authors illustrated that health research funders and regulators are promoting patient and public involvement in research, but this appears difficult to obtain because there is a lack of quality tools for involving patients.

Response:

Thank you for your comment. Our goal is to present the importance of patient and public involvement (PPI) in research and the main limitations in the patient and public engagement process.

2.     The Authors sustained that the partnerships with patients are necessary to produce and promote quality research, and that without the active participation of women as stakeholders the societal benefits of research cannot be realised. The Authors identified in the Creation and develop of platforms and opportunities for public involvement in sexual and reproductive health research a key international objective. The Authors sustained also that the cooperation between healthcare professionals, academic institutions and the community is essential to promote quality research and significant development in women's health will be achieved when involving of citizens in the research process becomes standard. This commentary is within the remit of International Journal of Environmental Research and Public Health, section Women’s Health, and it should be revised before it can be accepted for publication.

Response:

Thank you for your comments. We have revised the commentaries according to the reviewer’s suggestions.

3.     This is a commentary and not an original study or research study and for this reason I have only some comments or considerations to report:

3.1   The first sentence in the abstract is the same of the introduction, I suggest rephrasing one of them, it is not common to see the same phrase in different sections.

Response:

Thank you for your suggestions. We have modified the first sentence in the introduction.

Changes: Page 1. Lines 32 and 33

Evidence-based sexual and reproductive health requires a global effort with political and societal actions.

3.2   The Authors correctly highlight that as regards sexual and reproductive health there are still many goals to obtain also in the research area, and the PPI may be a correct strategy. However, I suggest deepening the sections related: to the creation of new tools and platform; to the specific model proposed by the authors, and to the studies and programme that have included PPI. I also suggest to deeply illustrate both the clinical relevance and the goals that should be obtained through this approach.

Response:

Thank you for your suggestions.

Changes: Page 2. Lines 61-66.

In this sense, creation of new tools and platforms involving a combination of traditional outreach and online strategies. Face-to-face invitations, training meetings, local civil society organizations (CSO) and family medicine health center support can be combined with online tools, such as online announcements, use of google meet, hangout or zoom, and social media platforms (Facebook, Twitter). These tools offer opportunities for community members to engage in interactive ways and can bring new input from the patient and participant involvement process.’.

We have expanded the section of the specific model proposed by the authors:

Changes: Page 6. Lines 243-258.

We suggest that these developments include clinicians (e.g. gynaecologists and paediatricians), methodologists (e.g. statisticians and systematic reviewers) and lay consumers (e.g. patients with lived experiences and civil society organization – CSO – representatives and related to sexual and reproductive health). Clinicians and professional bodies should promote patient and public participation in research helping to develop know-how in participants engagement. Methodologists – researchers, guideline makers and systematic reviewers included–, will help in the scientific training of lay research enablers, to develop the local Know- how on participant involvement in research. Lay consumers will provide the voice of lay sexual and reproductive health participants of all background to engage in the research process and to be involved in the dissemination of the research. The patient perspective is valuable to ensure quality in healthcare. However, the patient perspective is only one perspective, similar to the ones of other important groups (e.g., health professionals, leaders, administrators, politicians). Selected coordinator for the patient and public involvement is key to facility interaction. We agree with other authors, the coordinator should be experienced in the potential challenges of public and patient involvement, including power and control issues, and the consequences of public and patient contributors lacking knowledge of research processes, terminology, and ethical constraints [44].

We have added a section with studies that have included PPI in research and clinical relevance and goals:

Changes: Page 8 and 9. Lines 340-392

The most documented beneficial of patient and public involvement in research have been the involvement of the patient and public group at all stages of the trial [44], utility in randomized controlled trials [2], contribution in the design of clinical trials [43], and perceptions on patient involvement in research and clinical practice [1] .

In a qualitative study of patients and researchers from a cohort of randomized trials [2], 21 chief investigators, 10 trial managers and 17 patient and public involvement contributors were interviewed from 28 trials. Conclusions were that to enhance patient and public involvement in trials is crucial to develop goals for patient and public involvement at an early stage that fits the needs of the trial, plan patient and public involvement implementation in accordance with these goals, invest in developing good relationships between patient and public involvement contributors and researchers, and favour responsive and managerial roles for contributors. A systematic overview of public and patient involvement in clinical trials design included 27 reviews [43]. Public involvement roles were primarily in agenda setting, steering committees, ethical review, protocol development, and piloting. Public involvement was reported to have increased the quantity and quality of patient relevant priorities and outcomes, enrolment, funding, design, implementation, and dissemination.

We would like to mention a couple of examples were patient and public were involved in research. The first is the 3D study [44], were fourteen people living with multiple long-term conditions (multimorbidity) were patient and public involvement contributors to a randomised controlled trial to improve care for people with multimorbidity. Meetings took place approximately four times a year throughout the trial, beginning at grant application phase. Their experience also supports the findings of the RAPPORT study [57] in four of the six conditions most influential in establishing effective patient and public involvement: (1) a research team positive about patient and public involvement, (2) a key individual to co-ordinate, ensuring diversity, and relationships that are established and maintained over time, (3) a shared understanding of moral and methodological purposes of patient and public involvement and (4) proactive and systematic evaluation of patient and public involvement. The second example is a secondary analysis of women's perceptions and experiences of egg aspiration in fertility treatment [1]. This secondary analysis was initially inserted in a randomized controlled study where the aim was to gain insight into perceptions and experiences within a group of women undergoing fertility treatment through two focus group interviews. They pointed out that knowledge and awareness of the difference in perspectives is important when healthcare professionals seek to involve patients both in clinical practice and in research.

Then, patient and public involvement on research has double impact in trials: focussed and diffuse [2]. Focussed impact comprised patient and public contributors’ input that, from the perspective of the informant, changed or influenced an aspect of the trial, whereas diffuse impact comprised patient and public involvement contributions that influenced the way researchers thought or felt about the trial. Focussed impact included patient and public involvement contributors helping to choose the primary outcome for the trial and to increase recruitment through their contacts and networks. Diffuse impact largely entailed interactions between researchers and patient and public involvement contributors that helped to reassure the research team and increase or maintain their confidence and motivation for the trial.

Specific examples of impact of patient and public involvement in their professional life are those from patients becoming co-author in publications, co-applicants in projects call and trainers [4, 58-60] or even becoming new editor in a prestigious journal as the BMJ Sexual & Reproductive Health [61], who wrote: ‘Patient involvement is an important piece of cultural change in healthcare, and will require commitment, time and sensitivity, and a willingness to experiment and learn from mistakes’.

4.     Check for English language.

Response: We have checked the English level throughout the manuscript with a native language speaker.

Reviewer 2 Report

Above all, it is an opinion paper and not so much a bibliographic review.

To be a bibliographic review, the text should include divergent opinions.

It is not enough to call attention and highlight the advantages associated with the involvement of the general population in research and in the definition of health care policies and organization.

It is necessary to draw attention to the inconveniences that arise from this option and have the courage to analyze and criticize them.

In addition, the authors do not address the levels of knowledge that people have about health issues. The lack of health literacy is still a worrying reality in most democratic societies.

However, many clinicians, researchers, systematic reviewers and guideline makers still don’t have the tools required to involve patients. It is not always recognized that through their lived experiences and social background, patients can positively be involved in research with their unique perspectives, increasing its impact on society

Author Response

Commentary or suggestion

Author´s responses and changes

1.     Above all, it is an opinion paper and not so much a bibliographic review. To be a bibliographic review, the text should include divergent opinions.

Response:

Thank you for your comment. This is a commentary paper, and it has been well referenced (61 citations). We have extracted information from different experiences and approaches in the field of citizen participation in research.

2.     It is not enough to call attention and highlight the advantages associated with the involvement of the general population in research and in the definition of health care policies and organization.

It is necessary to draw attention to the inconveniences that arise from this option and have the courage to analyse and criticize them.

Response:

Thank you for your comments. We have added text to give consideration to how inconveniences have to be balanced against the risk of doing research without patient and public engagement. Also, we have included weaknesses of patient and public involvement in research.

Changes: Page 1 and 2. Lines 33-46

Evidence-based sexual and reproductive health requires a global effort with political and societal actions. A comprehensive paradigm shift is needed where not only health professionals and researchers are ready to implement citizen participation, but also where political and administrative control of the health system allows and supports it [1] . Public and patient involvement is a field where policy has tended to outpace evidence. Exploring the impact of citizens in research it is limited to investigating researchers’ and public and patient contributors’ reports of their views and experiences. Objective techniques for evaluating impact and its influences remain hard to reach in a process that is inherently relational, subjective and socially constructed [2]. Public and patient involvement is as an expression of a democratization of healthcare, and a political and managerial tool to ensure quality, documentation, and equal treatment, thereby governing and controlling a public healthcare system. For example, The Danish public healthcare system is increasingly influenced by politicians. The use of standardized schemes and checklists used for documentation and quality assurance underlines this development, and patient and public involvement might be seen as adding to this movement [1].

Changes: Page 4. Lines 172-184

Involving lay volunteers for problem‐solving provided insights enhanced research design and served to identify weaknesses and barriers. Tensions and barriers have been generated with patient and public involvement in research [43]. In one hand, shared tensions usually are due to unclear roles, absent reporting guidelines, exclusion, framework limitations, resource allocation, and administrative boundaries. For example: active public involvement in the decision‐making process of designing trials is less common than consultation on what was already decided. Volunteers report needing early involvement to propose constructive changes. Researchers are worries about aggressive patients and those without respect for rules of confidentiality or data protection harmed the research. Tension can occur over who has control of the research and have the perception that patient and public involvement contributors wanted more influence over the research than researchers felt they could give [44]. On the other hand, shared barriers include those imposed by cultures, values, and power hierarchies. Limited involvement of the health community may occur through coalitions, collaborations, and partnerships [43].

3.     In addition, the authors do not address the levels of knowledge that people have about health issues. The lack of health literacy is still a worrying reality in most democratic societies.

Response:

Patients are people with their lived experience of the disease and citizens are people with their own life experience. They can contribute with their perspectives on health and disease in general terms and in sexual and reproductive health specifically.

Changes: Page 4 and 5. Lines 184-197

Because research literacy could be an inconvenience in the process of patients and public involvement, some suggestions for getting the best from public involvement has been published [43]: ongoing support and implementation; training/capacity building; inclusion process; building trust and community and reinforce value and validate. In the second suggestion, capacity building, it is necessary to (1) provide training in research literacy and ethics, drawing on the many training programs that are available, (2) at every meeting have a jargon bin, when an unfamiliar term comes up, define it and use this to build glossaries, (3) promote a reciprocal learning relationship, letting volunteers know that researchers have made a long‐term commitment to patient and public partnership in research, and (4) encourage realistic expectations in volunteers and researchers and manage relationships with respect. Moreover, the use of patient and public involvement in dissemination planning, design, implementation, and distribution could increase public involvement, contribute to health literacy, and expand knowledge for patient values and preferences. The addition of patient reviewers by journals may contribute to health literacy and provide insights for future participatory research practice [43].

4.     However, many clinicians, researchers, systematic reviewers and guideline makers still don’t have the tools required to involve patients. It is not always recognized that through their lived experiences and social background, patients can positively be involved in research with their unique perspectives, increasing its impact on society.

Response:

We agree with the reviewer. We have noted that the health research process requires a combination of health professionals, researchers, and patient and public participation. Without citizen participation, research investigations are incomplete.

Round 2

Reviewer 2 Report

The paper corresponds to an innovative work proposal to facilitate the involvement of public opinion and patients in the life and functioning of health systems.